# High Pressure Processing Applications in Plant Foods

**DOI:** 10.3390/foods11020223

**Published:** 2022-01-14

**Authors:** Milan Houška, Filipa Vinagre Marques Silva, Roman Buckow, Netsanet Shiferaw Terefe, Carole Tonello

**Affiliations:** 1Food Research Institute Prague, 102 00 Prague, Czech Republic; 2LEAF, Linking Landscape, Environment, Agriculture and Food, Associated Laboratory TERRA, Instituto Superior de Agronomia, Universidade de Lisboa, Tapada da Ajuda, 1349-017 Lisboa, Portugal; fvsilva@isa.ulisboa.pt; 3Department of Chemical Engineering, University of Riau, Pekanbaru 28293, Indonesia; evelyn@eng.unri.ac.id; 4Faculty of Engineering, School of Chemical and Biomolecular Engineering, The University of Sydney, Darlington, NSW 2006, Australia; roman.buckow@sydney.edu.au; 5CSIRO Agriculture and Food, Werribee, VIC 3030, Australia; Netsanet.Shiferawterefe@csiro.au; 6Hiperbaric, S. A., Condado de Trevino, 6, 09001 Burgos, Spain; c.tonello@hiperbaric.com

**Keywords:** fruit and vegetable products, HPP, heat assisted HPP, effect on microorganisms, effect on endogenous enzymes, industrial equipment

## Abstract

High pressure processing (HPP) is a cold pasteurization technology by which products, prepacked in their final package, are introduced to a vessel and subjected to a high level of isostatic pressure (300–600 MPa). High-pressure treatment of fruit, vegetable and fresh herb homogenate products offers us nearly fresh products in regard to sensorial and nutritional quality of original raw materials, representing relatively stable and safe source of nutrients, vitamins, minerals and health effective components. Such components can play an important role as a preventive tool against the start of illnesses, namely in the elderly. An overview of several food HPP products, namely of fruit and vegetable origin, marketed successfully around the world is presented. Effects of HPP and HPP plus heat on key spoilage and pathogenic microorganisms, including the resistant spore form and fruit/vegetable endogenous enzymes are reviewed, including the effect on the product quality. Part of the paper is devoted to the industrial equipment available for factories manufacturing HPP treated products.

## 1. Introduction

High pressure processing (HPP) is a cold pasteurization technology by which products, prepacked in their final package, are introduced to a vessel and subjected to a high level of isostatic pressure (300–600 MPa). Although first studies of application of HPP technology to preserve milk date back to 1899 [1], the commercial use only started in the 1990s. HPP offers several advantages, including safe and minimally-processed products with extended shelf-life, improvement to the supply chain operations, food waste reduction, clean labels, versatility, eco-friendliness and a wide range of applications. HPP alone does not inactivate most microbial spores and enzymes; therefore, the combination of high pressure processing with heat, also referred to as high pressure thermal processing (HPTP) is required [2].

High-pressure treatment of fruit, vegetable and fresh herb homogenate products offer nearly fresh products in regard to the sensorial and nutritional quality of original raw materials representing relatively stable and safe source of nutrients, vitamins, minerals and health effective components. Such components can play an important role as a preventive tool against the start of illnesses, namely in the elderly. There is a need to provide reliable overview of recent scientific results related to HPP technology.

This review is divided into several sections, which deal with selected foods in the market network, the effect of high pressure treatment alone or in combination with heating on pathogenic and spoilage causing microorganisms, the effect on enzymes in fruits and vegetables and the history of industrial-scale equipment. This section contains a description of the novelty of how to use the volume of the pressure chamber as efficiently as possible. Parts of this review paper are based on upgraded content of chapters presented in a book edited by Houska and Silva [2,3,4].

An important goal of our work was not only scientific but also educational to provide those interested in this technology with all aspects important for future application in practice.

## 2. Food Products of Fruit and Vegetable Origin

Due to consumer demand on minimally processed food, the gentle preservation technologies like HPP at cold conditions are welcomed. There are several HPP products available on the market, including fresh meat, ham, oysters, and juices. The practice of HPP of raw cold prepared fruit and vegetable juices and smoothies is growing fast, because the treatment largely retains the original raw character of juices and quality, promotes safety and prolongs shelf life.

### 2.1. Juices and Smoothies Treated by HPP at Cold Conditions

An overview of products and companies around the world, including links on their web sites, is shown in Table A1, Table A2, Table A3, Table A4, Table A5, Table A6, Table A7, Table A8, Table A9, Table A10, Table A11, Table A12, Table A13, Table A14, Table A15 and Table A16 (Appendix A). Most of the juices/smoothies on the market are prepared with several different fruits and/or vegetables, and fewer as a single sort. Orange, lime or grape juices are the most frequent single sort types. There are interesting products such as HPP guacamole, an avocado mash of green color, whereby the traditional heat process causes browning, as opposed to HPP (see Table A1), which retains the original avocado green color. There are other interesting products such as juices or smoothies containing ginger or spirulina. It has been demonstrated that this modern technology retains important health active components of the raw fruits/vegetables.

### 2.2. Concluding Remarks

In the aforementioned tables, high value commercial products prepared from bio or organic produce by high pressure processing technology are presented. Such products are increasingly more welcomed by consumers in countries that can afford such technology, which is relatively expensive from an investment point of view. We hope that the attractive colorful spectrum shown in the tables can illustrate the commercial products currently available.

## 3. Effects of High Pressure Processing Combined with Heat (Further HPTP) on Key Microorganisms in Fruit and Vegetable Products

Silva and Evelyn [2] published results of effects of HPP and HPP combined with heat treatment on microbial inactivation in fruit and vegetable products (e.g., juices). Food authorities recommend six log reduction pasteurization in the most relevant pathogen, ranging from four to eight log reductions, depending on the food and the microorganism. HPP can damage the microbial cell membrane, which affects its permeability and ion exchange, and denature proteins involved in microbial replication. After pasteurization the fruit and vegetable products may contain microorganisms in lower concentration. Therefore, they are cold stored and distributed at temperatures below 7 °C to avoid or retard undesirable microbial growth during storage. The low temperature also inhibits enzymatic or other biochemical spoilage reactions. Spores are a resistant form of microorganisms that only specific microbial species can produce under adverse conditions. They are crucial given their resistance to pasteurization and often being able to survive and grow in the food after the process, thereby causing spoilage. For spores, high pressure thermal processing or HPTP is recommended by Evelyn and Silva [5]. By combining high pressure and temperature, we can achieve a synergistic effect on the microorganisms present. Given spore resistance to HPP and possible growth at refrigerated storage/distribution conditions; Silva and Evelyn [6] recommend the use of molds as pasteurization targets for cold distributed high pressure and heat assisted high pressure processed fruit products. Additionally, a study demonstrated that aged mold spores were more resistant to HPP, as opposed to bacterial spores [7]. HPP processors also have to demonstrate the process conditions inactivate pathogens of concern in a specific food. For example, it is known that *Salmonella* and *Escherichia coli 0157:H7* possess relatively high resistance to acidic environments found in fruit products, being able to survive up to several weeks at pH ≤ 4.6. Although growth is not probable in this acidic environment, their low infectious dose (10–100 cells) can become a public health concern even in the absence of growth, according to the FDA [8]. In addition to fruit beverages, HPP technology has been successfully used to pasteurize milk [9], beer [10] and wine [11]; these latter studies on yeast inactivation in beer and wine.

### 3.1. Modeling the Inactivation of Microorganisms in Fruit Products

A review of the models and its parameters for microbial inactivation in fruit products is presented in Table 1. In general, the inactivation of microbial vegetative cells in fruits was linear, described by a first order kinetics model, with parameters *D*- and *z*-values. On the contrary, the spores’ inactivation in fruit products exhibited a non-linear behavior with an upward concavity and tail. For non-linear inactivation the Weibull model was used (*log N*/*N*_0_ = −*b* × *t^n^*), where *b* and *n* parameters are the Weibull scale and shape factors, *log N*/*N*_0_ is the microbial log reductions, and *t* the HPP processing time (during holding phase) in minutes. Other authors used the first-order biphasic model, which assumes two rates of inactivation corresponding to two *D*-values.

The increase of spore resistance with processing time can result in tails in the survival curves and pose a problem to HPP processors, it being more difficult to inactivate the spores as processing time increases. The increase of the process pressure and/or temperature is a way to increase microbial inactivation. However, most of the commercial machines operate at a maximum pressure of 600 MPa and room temperature non-thermal conditions (maximum temperature after compression below 45 °C), although for experimental purposes, Hiperbaric built in 2009 a unique piece of equipment that enables the combination of pressure (630 MPa working pressure) and temperature (5–90 °C).

For *Alicyclobacillus acidoterrestris* spores, at 600 MPa for apple juice the *D*_45°C_-value = 8.6 min, Uchida and Silva [12]. As expected, the *D_600MPa,_*_45°C_-value increased from 8.6 min in 10.6°Brix apple juice to 20 min in 20°Brix lime juice concentrate and 46 min in 30°Brix blackcurrant juice concentrate. First order kinetics parameters were also determined for *Saccharomyces cerevisiae* ascospores in juices, which were much easier to inactivate at ≤40 °C (*D*_500MPa_-value 0.07 min), Parish [18]. With respect of spores of *Bacillus coagulans*, and molds *Byssochlamys nivea* and *Neosartorya*
*fischeri*, the non-linearity with concave upward (*n* between 0.35–0.68) was best described by the Weibull model, Weibull [22]; Daryaei and Balasubramaniam [13]; Evelyn and Silva [15]; Evelyn et al. [16]. One other option for these sort of non-linear survival curves is the first order biphasic, which divides the inactivation into two straight lines, corresponding to two different rates, the first higher inactivation rate presenting lower *D*-value, followed by a more resistant microbial population with higher *D*-value. The biphasic model was fitted to *B. coagulans* (600 MPa-60 °C, *D*-values of 1.6 and 6.2 min), and mold *Eurotium repens* (500 MPa-45 °C, *D*-values of 2 and 9 min), Merkulow et al. [17]; Zimmermann et al. [14].

Regarding vegetative cell inactivation in fruit products, log linearity with HPP processing time was registered and first order kinetic parameters were estimated. To be able to obtain survival data and model the kinetics low HPP pressures were used as otherwise the inactivation would be too quick (in seconds). For example, at 350 MPa, the *D*-value ranged between 0.6 and 2 min. The *D*-value for *Lactobacillus brevis* bacterium and *Zygosaccharomyces bailii* yeast in orange juice was 0.6 min, Katsaros et al. [20]; Hiremath and Ramaswamy [21]. Similar to bacterial spores, the *Saccharomyces cerevisiae* vegetative cells increased resistance from single strength orange juice to 42°Brix concentrate, Basak et al. [19]. The *D*-values at 600 MPa can be estimated from the *z_P_*-value provided by the authors. Those are very useful to estimate minimum processing conditions equivalent to 6D or 6 log reductions. The estimated *D*_600MPa_-values for *Leuconostoc mesenteroides* in orange juice and orange juice concentrate are 2 s and 58 s, respectively. *D*_600MPa_-values in orange juice of *Lactobacillus brevis* is 0.12 s.

### 3.2. Modeling the Inactivation of Microorganisms in Vegetable Products

Vegetable products such as soup, some sauces, food ingredients and certain fruit juices (e.g., tomato, pear, some tropical juices) are examples of low-acid foods, as their pH > 4.6. Vegetative bacterial pathogens (e.g., *Salmonella*, *Escherichia coli*), bacterial spores from pathogens (*Clostridium botulinum*, *Bacillus cereus*) can grow at pH > 4.6 and pose a risk to human health. Bacterial spores from spoilage species (*Geobacillus stearothermophilus* = *Bacillus stearothermophilus*) can also grow. To minimize the outgrowth of pathogenic microbes in the foods during distribution, the microbial spores surviving pasteurization must be controlled, by using cold storage and transportation (1 to 8 °C), and a limited shelf-life [23]. For this class of foods, food processors have to demonstrate that the processed food is safe, not capable of supporting the growth and toxin production by *C. botulinum* within the specified storage life of the food. HPP process itself can reduce slightly the pH of beverages. In practice the pH can be lowered to safe levels by combining vegetable juices with fruit juices which contain high concentrations of acids [24].

Table 2 shows the modeling of microbial inactivation carried out in vegetable products. *Bacillus licheniformis* spore inactivation in carrot juice and *Eurotium repens*/*Penicillium expansum* spore inactivation in broccoli juice were non-linear. The first was modeled with Weibull distribution [25] and the second used a biphasic model. For *E. repens* submitted to 500 MPa-45 °C, initially the *D*-value was 16 min and after 125 min a plateau without any inactivation was registered. Regarding *P. expansum*, first the *D*-value was <1 min followed by no inactivation (350 MPa-40 °C) [17]. Although *B. licheniformis* inactivation was non-linear for 400–500 MPa and 40–50 °C, it was readily inactivated at 600 MPa combined with the higher temperature tested of 60 °C, and the inactivation in this case was close to linear. Therefore, the first order model was also used successfully with a *D*-value of 0.70 min for 600 MPa-60 °C [25]. The results demonstrated how important it is to use higher pressures and temperatures to obtain a quick inactivation of the microorganisms. The inactivation of the vegetative microorganism *E. coli* in carrot juice did not require heat and was log linear presenting a *D*-value of 2.5 min at 600 MPa [26].

### 3.3. Concluding Remarks

In general, 600 MPa HPP at room temperature successfully inactivated microbial vegetative cells in fruit and vegetable products, including spoilage and pathogenic organisms (e.g., *Escherichia coli*, *Listeria monocytogenes*). The inactivation of vegetative cells followed a linear pattern described by a first order kinetics model, with parameters *D*- and *z*-values. Yeast spores were also inactivated at room temperature HPP. On the contrary, heat assisted high pressure processing (HPTP) with process temperatures above 60 °C were required for bacterial and mold spore inactivation. In this case, the inactivation exhibited a non-linear behavior with an upward concavity and tail. The Weibull or the first order biphasic models were used by most of the authors to describe the non-linear inactivation pattern. As commercial HPP units usually do not operate with supplied heat, HPP products are generally distributed and stored under refrigeration (<7 °C), as this low temperature will avoid the germination and growth of spores (e.g., *Clostridium botulinum*).

## 4. The Effect of High Pressure on Endogenous Fruit and Vegetable Enzymes

### 4.1. Introduction

One of the main challenges in the shelf-life extension of fruit and vegetable products by HPP is the control of endogenous enzymes, which often exhibit a higher resistance to high pressure conditions than vegetative spoilage bacteria [27]. Therefore, this paper partly focuses on the challenges in the application of HPP for modulating endogenous enzyme activity in a wide range of horticultural products (see Chapter 3 in [4]). Enzymes are biocatalysts that are essential for the physiological and metabolic activities of living organisms. As such plants synthesize various enzymes at the different stages of their developments some of which remain active after harvesting. This is essential in cases where ripening takes place during postharvest storage (e.g., ripening of climacteric fruits such as banana). However, the continuous activity of endogenous enzymes during postharvest processing and storage often leads to degradation in quality parameters such as color, flavor, texture and nutritional attributes. The shelf-life of horticultural products is significantly shortened by the activity of endogenous deteriorative enzymes, the growth of spoilage microorganisms (with associated enzymatic activity) and other non-enzymatic (usually oxidative) reactions [27]. Thus, one of the objectives of post-harvest processing for shelf-life extension of horticultural products is the control of deteriorative enzyme activity. These can be achieved in many ways including removal of substrate molecules (e.g., oxygen for oxidative enzymes), inhibition of enzyme activity (e.g., sequestration of metal ions in the active sites of enzymes), and enzyme denaturation through the application of physical or chemical agents (e.g., thermal inactivation).

The inactivation of enzymes occurs in general when their secondary and tertiary structures are disrupted. This disruption may be caused by different physical (e.g., heat, high pressure and ultrasound) and chemical stressors. Nevertheless, the mechanism of structural disruption and denaturation depends on the stressor. For instance, heat and chemical induced denaturation results in complete unfolding and irreversible denaturation of enzymes whereas high pressure induced denaturation can leave parts of the enzyme structure unchanged [28]. High pressure at 150 to 200 MPa causes the disruption of quaternary structure that results in the dissociation of oligomeric enzymes into individual subunits ([29,30]) as was observed in the case of β-lactoglobulin [29]. At higher pressures, the disruption of the tertiary structure of enzymes that is largely stabilized by electrostatic and hydrophobic interactions, takes place [31]. This is accompanied by penetration of water into the interior part of the protein molecule, which leads to loss of contact between non-polar domains of the molecule resulting in conformational change, partial unfolding and loss of activity [32,33,34], which was demonstrated through modeling [32] and experimental observations for 3-isopropylmalate dehydrogenase [33] and mitochondirial FoF1-ATPase [34]. Enzyme-substrate interaction occurs on a hydrophobic cleft known as the active site and even the slightest change in the structure of the active site may lead to a complete loss of enzyme activity [35].

### 4.2. Effects of High-Pressure Processing on Quality Degrading Enzymes in Fruits and Vegetables

As discussed above, the shelf-life of processed fruit and vegetable products is limited by the activity of endogenous deteriorative enzymes and other non-enzymatic (usually oxidative) reactions in addition to the growth of spoilage microorganisms [27]. Tissue and cell disruption during processing causes enzyme-substrate de-compartmentalization and interaction that leads to changes in color, flavor, texture, consistency and nutritional attributes. Thus, considerable effort has been devoted over the years for determining the high pressure processing conditions combined with other hurdles (temperature, pH etc.) for inactivation or inhibition of endogenous enzymes in high pressure processed fruit and vegetable products. The following is a summary of the key findings.

#### 4.2.1. Enzymes Related to Texture, Consistency and Cloud Stability

Some of the most important quality parameters that determine the acceptability of processed fruit and vegetable products are texture (for vegetable and fruit pieces/slices), consistency (purees, pastes, juices and nectars) and cloud stability (for non-clarified juices). These are largely determined by the structure and shape of individual cells, types of tissue (e.g., presence of strong vascular tissue), composition (type/proportion of polysaccharides) and the chemical and physical structure of cell wall polysaccharides [27], which are dependent on the product and the degree of processing it has undergone. The changes in texture and consistency of horticultural products during processing and storage are largely caused by enzymatic and non-enzymatic changes in the structure of pectin [36]. The main endogenous enzymes that are involved in pectin modification are pectin methylesterase (PME) and polygalacturonase (PG). PME catalyzes the de-esterification of pectin into low-methoxy pectin; an ideal substrate for the activity of PG, which catalyzes the depolymerization of low methoxy pectin resulting in loss of firmness in solid products [37] and decrease in viscosity and consistency and syneresis during storage in products such as tomato juice [38]. The PME catalyzed conversion of pectin into low-methoxy in the absence of active PG on the other hand results in better firmness and consistency since (1) demethylated pectin chains tend to cross-link in the presence of divalent cations (e.g., Ca^2+^) and form the so called ’egg box’ structure and (2) Demethylated pectin resists degradation during thermal process [27]. Another quality attribute that is affected by PME activity is juice cloud, which is stabilized by soluble pectin in the juice [39]. The free carboxyl groups in PME de-methylated pectin bind with divalent cations such as Ca^2+^ in the juice resulting in the formation of cross-linking with adjacent pectin chains that form aggregates and settle resulting in cloud loss [40,41]. Cloud loss in products such as orange juice is considered as a serious quality defect since it affects not only visual appeal but also flavor, aroma and color of the juice.

There are several studies in the scientific literature on the effects of high pressure and HPTP on the activity of PME and to a lesser extent PG in fruit and vegetable products. A summary of representative studies is presented in Table 3. Among the studies, many are not surprisingly on orange PME due to its role on the cloud stability of orange juice. In conventional orange juice processing, orange juice is thermally processed at 90–99 °C for 15–30 s with the aim of inactivating the thermoresistant isozyme of PME [42], since it is the isozyme mainly responsible for cloud loss [43]. Interestingly, the heat labile orange PME is also pressure labile and significant inactivation of this fraction has been reported after short treatment of orange juice at 500 MPa and higher (Table 3). Nevertheless, there is substantial variation in the reported effects, perhaps due to differences in variety and growing conditions among other factors. For instance, Vervoort [44] reported 92% inactivation of PME in orange juice (pH 4.0) made from three varieties (Valencia, Pera and Baladi) after HPP treatment at 600 MPa, 15 °C for 1 min whereas treatment Valencia orange juice (pH 4.3) after treatment at 600 MPa, 20 °C for 1 min [45]. The thermoresistant orange PME on the other hand appears to be pressure resistant with no complete inactivation at pressures as high as 900 MPa [46,47]. Despite the residual PME activity, HPP processed orange juice has been reported to maintain good cloud stability for two months and more under refrigerated and accelerated storage conditions [47,48]. Similarly, better cloud stability was reported for guava puree subjected to HPP (600 MPa, 25 °C, 15 min) despite the 76% residual PME activity compared to 4% residual PME activity in thermally treated (90 °C/24 s) puree [49]. The cloud stability of HPP treated products despite significant residual PME activity is attributed to the shear induced modification of the structure of pectin making it inaccessible to PME and a reduction in the size of the suspended cloud particles [27].

The other PME that is well investigated is tomato PME, which is one of the most pressure stable enzymes (see Table 3) which resists inactivation even after 5 min treatment at pressure as high as 800 MPa and 45 °C [50]. PMEs from other sources are also quite pressure stable, although increasing temperature or pressure hold time have been shown to increase the degree of inactivation (Table 3). PG, on the other hand, is one of the enzymes which is relatively sensitive to pressure induced inactivation although studies are limited to tomato PG. HPP treatment of tomato dices at 600 MPa and 25 °C for one minute caused 60% inactivation of tomato PG [50] whereas 15 min treatment of tomato pieces or juice at the same condition resulted in the complete inactivation of the two PG isozymes [51]. The difference in the relative pressure stability of PME and PG is advantageous in vegetables with endogenous PME and PG activity such as tomato since PG is selectively inactivated during HPP processing under commercially feasible condition (500–600 MPa) while PME remains active and its pectin de-methylation activity is enhanced under high pressure. This makes the formation of an egg-box structure in the presence of divalent cations possible, resulting in improved texture and consistency of high pressure processed products as described above.

#### 4.2.2. Enzymes Related to Color Stability

Color is an important quality attribute that influences the visual appeal and the acceptability of food products. The major pigment compounds in fruits and vegetables are polyphenols, including anthocyanins, chlorophyll, and carotenoids [52]. The degradation of these pigments not only affects color but also the nutritional and health benefits derived from the biological activity of some of the pigment compounds such as polyphenols and carotenoids. The main enzymes responsible for color degradation are polyphenol oxidase (PPO) and peroxidase (POD), which are responsible for enzymatic browning caused by the oxidation of polyphenols into quinones followed by polymerization into brownish pigments [53]. PPO and POD do not directly catalyze the oxidation of anthocyanins, the major pigment compounds responsible for the color of many fruits and vegetables, due to steric hindrance by their glycosidic moiety. The cleavage of the glycosidic group by the action of β-glucosidase makes anthocyanin aglycons susceptible to the actions of PPO and POD [54]. Another enzyme, lipoxygenase (LOX), catalyzes the co-oxidative degradation of carotenoids in the presence of free fatty acids [52] whereas PPO causes the co-oxidative degradation of carotenoids in the presence of polyphenols [55]. Chlorophyll may undergo both enzymatic and non-enzymatic degradation during processing. During thermal processing at low pH, chlorophyll undergoes acid hydrolysis into pheophytin, which is further converted to pheophorbide by chlorophyllase catalyzed or chemical cleavage of the phytol chain [56]. POD and LOX have also been reported to catalyze the co-oxidative degradation of chlorophyll in the presence of a polyphenol or linoleic acid respectively [57].

A summary of representative studies on impact of HPP on the major color degrading enzymes polyphenol oxidase (PPO) and peroxidase (POD) in various fruit and vegetable matrices are presented in Table 4. PPO is one of the most pressure resistant enzymes often requiring pressure in excess of 600 MPa for a measurable inactivation at around room temperature within reasonable treatment time (<15 min) (Table 4). Nevertheless, the stability of PPO towards high pressure inactivation is dependent on the source (species/cultivar), the processing temperature and the physicochemical properties of the matrix (pH, Brix). For instance, no inactivation of PPO was observed in Royal Gala apple puree subjected to 600 MPa at 34 °C for 5 min [58], Taylor’s Gold pear puree subjected to 600 MPa, 34 °C for 5 min [59], strawberry (cv. Festival) halves treated at pressures up to 600 MPa at 60 °C and 10 min and strawberry (cv. Aroma) puree subjected to pressure treatment at 690 MPa and 24 °C for 15 min. On the other hand, a significant level of PPO inactivation has been reported under comparable conditions in many cases, including strawberry pulp (unspecified cultivar) [60], nectarine puree [61], watermelon juice [62] and carrot juice [63]. Factors such as treatment pressure and temperature, acidity and brix have significant effect on the resistance of PPO to inactivation. For instance, reducing the pH of guava juice from 4.7 to 3.9 increased the level of PPO inactivation after high pressure treatment (600 MPa, 25 °C, 10 min) from 45% to 70%, whereas increasing the sugar content from 3 to 12 Brix at the same pH (3.9) reduced the level of inactivation from 70% to 50% under the same processing condition [64]. In contrast, an increase in the degree of PPO inactivation was observed with increase in sugar content in strawberry puree subjected to high pressure treatment at 200 to 600 MPa and 40 to 80 °C for 2.5 to 10 min. In many cases, PPO activity increase is observed after high pressure treatment [65,66,67,68] at ambient to mild temperature conditions (Table 4). This is attributed to high pressure induced tissue disruption and release of membrane bound enzymes as well as activation of the latent form of PPO due to pressure induced change in protein structure [27,69].

POD is another enzyme, which is relatively resistant towards pressure induced inactivation. The relative sensitivity of POD and PPO varies depending on the matrix. In a study on strawberry halves (cv. Festival), no inactivation of PPO was observed after high pressure processing at 100 to 600 MPa and 20 to 60 °C for 2 to 10 min, whereas a maximum of 58% inactivation of POD was observed after treatment at 600 MPa and 60 °C for 10 min [70]. In contrast, Garcia-Palazon reported complete inactivation of PPO after 15 min treatment of strawberries at 600 MPa and 800 MPa, while only 11 to 13% inactivation of POD was observed under the same condition [67]. As other enzymes, the susceptibility of POD to high pressure inactivation is affected by factors such as matrix pH, sugar concentration and processing temperature. The high pressure induced inactivation of POD in Guava juice after treatment at 600 MPa and 25 °C for 10 min was 20% at pH 4.7 and increased to 50% at pH 3.9. The inactivation level decreased to 30% when the sugar content was increased from 3 to 20 Brix at the same pH (3.7) [64]. Increasing the treatment temperature at constant pressure often leads to increase in level of inactivation, although that depends on the enzyme and the pressure-temperature range, since application of moderate pressure at elevated temperature stabilizes many enzymes against thermal denaturation. For instance, with Packham pear slices in acidified syrup, high pressure processing (600 MPa, 5 min) at 20 °C resulted in 26% inactivation of PPO. Increasing the processing temperature to 40 or 60 °C resulted in 6% and 4% increase in PPO activity whereas further increase in temperature to 80 °C and 100 °C resulted in 79% and 92% inactivation respectively [71]. As in the cases of PPO, an increase in POD activity is commonly observed after high pressure processing [72,73,74], which can be due to pressure induced release of membrane bound enzymes.

Among color degrading enzymes, β-glucosidase is the least investigated with respect to impact of high pressure processing. β-glucosidase from strawberries exhibited 49% and 61% inactivation after 15 min processing at 600 MPa and 800 MPa whereas ~10% inactivation of β-glucosidase was observed after 15 min treatment of red raspberries at 600 and 800 MPa.

#### 4.2.3. Enzymes Related to Flavor Stability

Lipoxygenase (LOX) is one of the main enzymes responsible for off-flavor development in fruit and vegetable products [75]. The LOX catalyzed oxidation of polyunsaturated fatty acids and esters into the corresponding hydroperoxides and their further degradation into volatile compounds such as aldehydes, ketones and alcohols lead to off-flavor development often characterized as hay-like [75,76]. The activity of LOX also plays a significant role in the development of the desirable ‘fresh’ and ‘green’ flavor note in fruits and vegetables. LOX catalyzes the oxidation of linoleic and linolenic acids into their respective 13-hydroperoxides, followed by a selective cleavage by hydroperoxide lyase resulting in the formation of hexanals and hexenels from the 13-hydroperoxides of linoleic and linolenic acids respectively, which are the major volatile compounds responsible for the ‘fresh’ flavor of fruits and vegetables [77]. Polyphenolic compounds significantly contribute to the bitter, sweet, pungent or astringent taste of fruits and vegetables. Thus, the PPO and POD catalyzed oxidation of polyphenols may also impact the flavor and aroma of fruits and vegetables [78].

There are a number of studies in the literature on the impact of HPP on LOX mainly from vegetables, perhaps due to the importance of LOX inactivation for flavor stability of vegetables. In general, LOX is relatively more pressure labile compared to other plant enzymes, with substantial inactivation reported at 500 MPa and higher (Table 5). Most of the studies indicate synergistic inactivation effect of moderately high temperature (50 to 70 °C) with pressure at 500 MPa and higher [63,79,80], whereas antagonistic effects were reported at lower temperatures (≤10 °C) in some cases [79,80]. Interestingly, LOX was found to be more susceptible to pressure inactivation in whole green peas and green beans compared to the respective juices [79,80], which is somewhat counterintuitive.

#### 4.2.4. Enzymes Related to Nutritional Quality

The activities of endogenous enzymes affect the nutritional quality of fruits and vegetables directly or indirectly through co-oxidation mechanisms. For instance, the activity of LOX causes the degradation essential fatty acids such as linoleic, linolenic and arachidonic acids and indirectly leads to the degradation of carotenoids, tocopherols (vitamin E), and other nutrients through a co-oxidation in the presence of polyunsaturated fatty acids. The activities of PPO and POD lead to the degradation of polyphenols and carotenoids through co-oxidative mechanism in the presence of polyphenols. The endogenous enzyme myrosinase on the other hand enhances the nutritional quality of brassica vegetables since it catalyzes the conversion of glucosinolates into the bioactive isothiocyanates, which have been shown to have several health benefits [81].

Some studies investigated the impact of high pressure processing on broccoli myrosinase in broccoli pieces and broccoli juice. The enzyme was found to be quite pressure labile, although mild pressure (<200 MPa) stabilizes it against thermal inactivation [81,82], which can potentially be exploited to enhance the myrosinase catalyzed conversion of glucosinolates to isothiocyanates. Apart from its effects on myrosinase, high pressure may enhance the conversion of glucosinolates through its effect on the protein co-factor epithiospecifier protein, release of bound glucosinolates and enhanced enzyme-substrate interaction due to tissue disruption. A positive effect of HPP on the production of isothiocyanates has been reported for broccoli sprouts [83] and white cabbage [84].

**Table 3 foods-11-00223-t003:** Summary of relevant studies on the effect of HPP on pectin modifying enzymes in fruits and vegetables.

Product	Conditions and Enzyme Investigated	Results	Reference
Orange juice (pH 3.45)	PME 500 to 900 MPa, 20–50 °C not controlled, temperature effect not accounted for, *t* = 1 s	10 to 93% inactivation respectively after 1 s at 600 to 900 MPa (only the labile isoenzyme), slower inactivation at 500 MPa	[46]
Florida orange juice	PME 500–800 MPa, 25–50 °C, *t* = 1 min	800 MPa and 25 °C for 1 min reduced residual PME activity to 4% and good cloud stability over 2 months observed	[47]
Tomato dices	PME, PG 400, 600, 800 MPa, 25 and 45 °C, 1–5 min	No inactivation of PME after up to 5 min treatment at all conditions, 50% activation after 5 min at 400 MPa and 45 °C; 60% inactivation of PG at 600 MPa, 25 °C, 1 min and complete inactivation at 800 MPa, 25 °C, 1 min	[50]
Tomato juice Tomato pieces	PG 600 MPa, 25 °C, 15 min	Complete inactivation of the two isozymes of PG	[51]
Atemoya puree (pH 4.5)	PME, PG 100, 300, and 600 MPa, ~30–40 °C not controlled, 15 min	PG was inactivated by 65% and 82% at 300 and 600 MPa, respectively; PME was inactivated by 22% and 43% at 300 and 600 MPa, respectively	[85]
Greek Navel Orange juice	PME 100–800 MPa, 30–60 °C, kinetic	Inactivation of the labile form at all conditions including 100 MPa and 30 °C, antagonistic effect at low pressure (100–250 MPa) and higher temperatures (60 °C)	[86]
Valencia (pH 4.3) and Navel orange (pH 3.7) juice	PME 600 MPa, ~20 °C, 1 min	Treatment resulted in approximately 40% reduction in PME activity in Navel orange juice; no significant PME reduction was observed in HPP Valencia orange juice.	[45]
Mixture of Valencia, Pera and Baladi orange juices (pH 4.0)	PME 600 MPa, ~15 °C, 1 min	Approximately 92% inactivation	[44]
Greek Valencia variety (pH 3.8, 11.6°Brix)	PME 100–500 MPa, 20–40 °C, up to 30 min	5% inactivation after 20 min at 100 MPa and 30 °C; 85% inactivation after 1 min at 500 MPa and 30 °C	[20]
Cloudy apple juice Golden delicious	PME 200–600 MPa, 15–65 °C, 0.5–10.5 min	Activity increase up to 40 °C at all conditions, some inactivation at higher temperatures	[87]
Packham pears	PME 600 MPa, 20–100 °C, 5 min	49% and 78% inactivation at 20 and 60 °C, respectively	[71]
Strawberry puree with added sugar (0–30%)	PME, PG 200–600 MPa, 40–80 °C, 2.5–10 min	Maximum inactivation of 80%, 67% for PG, PME respectively; increased rate of inactivation at higher pressure and added sugar content	[88]
Aloe Vera juice (pH 2.32–5.68)	PME 60–740 MPa, 3–40 min	A maximum inactivation of 30% at 736 MPa for 20 min at pH 4.0; ncreased activity at 200 MPa, a maximum of 11% increase after 30 min treatment	[89]
Apricot nectars With added sugar (10 °Brix) and citric acid (pH 3.7)	PME (300 MPa, 34 °C), (400 MPa, 37 °C), (500 MPa, 40 °C), 5–20 min	No inactivation of PME	[72]
Mango pulp pH 3.5, 4.0, 4.5; 15, 20, 25 °Brix	PME 400–600 MPa, 40–70 °C, 6–20 min	Higher enzyme inactivation at lower pH and Brix and higher pressure and temperature at fixed pH and Brix; a maximum of ~54% inactivation at the lowest pH and Brix and 60 °C and higher regardless of the pressure	[78]
Peach pulp	PME 100–800 MPa, 30–70 °C, kinetic	Synergistic inactivation of pressure and temperature except at 70 °C and 100 to 600 MPa; only ~6% inactivation at 600 MPa, 30 °C, 3 min and ~93% inactivation at 600 MPa, 70 °C, 3 min estimated based on the kinetic data	[90]
Peach juice with 0.02% ascorbic acid	PME 400–600 MPa, 25 °C, 5–25 min	18.8% and 50.4% inactivation of PME at 600 MPa for 5 and 25 min respectively	[91]
Pineapple puree (pH 3.0, 3.5, 4.0)	PME 100–600 MPa, 20–70 °C, 0–30 min	Inactivation rate increased with decrease in puree pH; estimated optimum inactivation condition for PME and maximum retention of Bromelain (BRM) were 600 MPa/60 °C treatment time of 9 min for pH 3.0 and 10 min for pH 3.5 and 4 with 74% inactivation of PME and 49% retention of BRM at pH 3.5	[92]
Watermelon juice	PME 200, 400, 600 MPa, 25–33 °C, 5–60 min	15% inactivation of PME after treatment at 600 MPa for 15 min	[62]
Carrot juice	PME 100–600 MPa, 25 °C, 10 min 300–500 MPa, 50–70 °C, 10 min	Increased activity at 500–600 MPa with maximum 28.8% activation of PME at 600 MPa; maximum 97.6% inactivation of PME at 500 MPa, 50 °C	[63]
Carrot pieces	PME 700 to 800 MPa, 10 and 40 °C, kinetic	No inactivation at 10 °C, inactivation at 800 MPa, 40 °C	[93]
Tomato pieces	PME 0.1–500 MPa, −26 to 20 °C, 13 min	No inactivation of PME in the whole temperature and pressure range	[94]
Tomato juice	PME 550–700 MPa, 25 °C, kinetic	No inactivation	[95]

**Table 4 foods-11-00223-t004:** Summary of relevant studies on the effect of HPP on color degrading enzymes.

Product	Conditions and Enzyme Investigated	Results	Reference
Packham pears slices in syrup (20 Brix) acidified with citric acid (pH = 3.27)	PPO 600 MPa, 20–100 °C, 5 min	69%, 48%, 41%, 68% and 90% inactivation at 20, 40, 60, 80 and 100 °C, respectively; HPP at 40 and 60 °C resulted in significant higher residual enzyme activity	[71]
Packham pears slices in syrup (20 Brix) acidified with citric acid (pH = 3.27)	POD 600 MPa, 20–100 °C, 5 min	26%, 79% and 92% inactivation of this enzyme at 20, 80 and 100 °C, respectively; 6% and 4% increase in activity at 40 and 60 °C	[71]
Strawberry puree with added sugar (0–30%)	PPO, 200–600 MPa, 40–80 °C, 2.5–10 min	Maximum inactivation of 50%; increased rate of inactivation at higher pressure and added sugar content	[88]
Apricot nectars with added sugar (10 °Brix) and citric acid (pH 3.7)	PPO, POD (300 MPa, 34 °C), (400 MPa, 37 °C), (500 MPa,40 °C), 5–20 min	A significant activation of PPO (20% increase at 500 MPa to 45% increase at 300 MPa), POD (10% increase at 300 MPa to 45% increase at 500 MPa) regardless of hold time	[72]
Mango pulp pH 3.5, 4.0, 4.5; 15, 20, 25 °Brix	PPO, POD, 400–600 MPa, 40–70 °C, 6–20 min	Higher enzyme inactivation at lower pH and Brix and with increase in pressure and temperature at fixed pH and Brix; PPO and POD showed similar sensitivity to pressure inactivation	[78]
Peach juice with 0.02% ascorbic acid	PPO 400–600 MPa, 25 °C, 5–25 min	45.1% and 81.2% inactivation of PPO at 600 MPa for 5 min and 25 min respectively; 7.3% increase in activity at 400 MPa for 5 min	[91]
Pineapple puree (pH 3.0, 3.5, 4.0)	PPO, POD, 100–600 MPa, 20–70 °C, 0–30 min	Increased inactivation rate with decrease in puree pH; estimated optimum inactivation condition for PPO and POD and retention of Bromelain (BRM) were 600 MPa/60 °C treatment time of 9 min for pH 3.0 and 10 min for pH 3.5 and 4.0 with 64% and 67% inactivation of PPO and POD respectively and 49% retention of BRM at pH 3.5	[92]
Watermelon juice	PPO, POD 200, 400, 600 MPa, 25–33 °C, 5–60 min	87.7% and 42.4% inactivation of PPO and POD respectively after treatment at 600 MPa for 15 min	[62]
Carrot juice	PPO		[63]
	100–600 MPa, 25 °C, 10 min	90% inactivation of PPO at 500 and 600 MPa at 25 °C	
	300–500 MPa, 50–70 °C, 10 min	>80% inactivation of PPO (P = 300–500 MPa, T = 50–70 °C)	
Carrot pieces	POD 100–500 MPa, −26 to 20 °C, 13 min	~ 50% inactivation of POD at 100–200 MPa and 500 MPa and 20 °C ~ 50% inactivation of POD at 100–200 MPa and −10 °C; limited or no inactivation at other conditions	[94]
Royal Gala apple puree	PPO 600 MPa, 34 °C, 0–60 min	90% activity increase after HPP for 5 min; approximately 10% activity reduction after HPP for 60 min	[58]
Taylor’s Gold pear puree	PPO 600 MPa, 34 °C, 0–60 min	30% activity increase after HPP for 5 min	[58]
Golden Delicious apple juice	430–570 MPa, 1–8 min, PPO	50% activity increase after HPP for 1 min irrespective of applied pressure	[59]
Strawberry pulp	PPO, POD, β-glucosidase, 400–600 MPa, ~25 °C, 5–25 min	PPO inactivation at all conditions with the highest inactivation of 51% at 600 MPa, 25 min; highest POD inactivation of 71.4% at 500 MPa, 25 min. At 600 MPa, the residual POD activity increased with increase in treatment time from 5 min to 10 min from 35.7% to 71.9%; 17% increase in β-glucosidase activity at 400 MPa, 25 min. ~15% and 41% inactivation at 500 MPa, 25 min and 600 MPa/25 min respectively	[60]
Nectarine puree Untreated Thermally blanched (80 °C, 40 s) Puree with 400 ppm ascorbic acid	PPO 400 MPa, 600 MPa, 5 min	~25% and ~60% inactivation at 400 MPa and 600 MPa in the untreated sample, ~60% inactivation at both 400 MPa and 600 MPa in the blanched samples, and slight activation at 400 MPa and ~50% inactivation at 600 MPa in samples with added ascorbic acid	[61]
30% Guava juice (pH 4.7, 3 °Brix)	PPO, POD, 600 MPa, 25 °C, 10 min	45% inactivation of PPO and 20% inactivation of POD	[64]
30% Guava juice (pH 3.9, 3 °Brix)	PPO, POD 600 MPa, 25 °C, 10 min	70% PPO and 50% POD inactivation	
30% Guava juice (pH 3.9, 12 °Brix)	PPO, POD 600 MPa, 25 °C, 10 min	50% PPO and 30% POD inactivation	
Cloudy apple juice (pH 3.8) from cv. Boskop	PPO 0.1–700 MPa, 20–80 °C, kinetic	65% activity increase at 400 MPa, 20 °C, 5 min; antagonistic effect at pressure <300 MPa and temperature ≥60 °C for pressure-temperature inactivation after initial PPO activation	[65]
Shredded broccoli	PPO, POD 210 MPa and −20 °C, 180 MPa and −16 °C (Pressure-shift freezing)	No inactivation of PPO and POD	[66]
Strawberries	PPO, POD, β-glucosidase, 400–800 MPa, 5–15 min	76% increase in β-glucosidase activity at 400 MPa and 15 min. 49% and 61% inactivation after 15 min at 600 MPa and 800 MPa respectively; complete inactivation of PPO after 15 min at all pressures and about 69% inactivation after 5 min 11–13% inactivation of POD after 15 min at 600 and 800 MPa and slight activation (13%) after 5 min at 400 MPa	[67]
Red Raspberries	PPO, β-glucosidase, 600–800 MPa, 5–15 min	54% and 42% PPO activity increase after 5 and 10 min at 600 MPa and 29% inactivation at 800 MPa for 15 min ~10% inactivation of β-glucosidase at 600 MPa and 800 MPa after 15 min	[67]
Muscadine grape juice Co-pigmented and non-copigmented	PPO 400, 550 MPa, 15 min	3 and 2.5 times increase in PPO activity in non-copigmented juices 60% and 20% increase in PPO activity in juices co-pigmented with rosemary and thyme respectively at both pressures	[68]
Strawberry halves (cv. Festival)	PPO, POD, 100–600 MPa, 20–60 °C, 2–10 min	No significant inactivation of PPO; a maximum of 58% inactivation of POD at 600 MPa, 60 °C and 10 min	[70]
Apple pieces	POD 600–1000 MPa, 20 °C, 15 and 30 min	Two-fold increase in activity at 600 MPa, ~40% inactivation at 1000 MPa, no effect of treatment time	[73]
Whole lychee	PPO, POD, 200–600 MPa, 20–60 °C, 10 and 20 min	Increased activity of POD at 200 MPa (3 and 2.5 times increase at 40 °C for 10 and 20 min respectively), no effect at 400 to 600 MPa and 20 to 40 °C, inactivation at 600 MPa and 60 °C with over 50% inactivation at 600 MPa, 60 °C for 20 min; limited inactivation or activation after 10 min at all conditions; a maximum of 90% inactivation of PPO at 600 MPa, 60 °C for 20 min	[74]
Mixture of Valencia, Pera and Baladi orange juices (pH 4.0)	POD 600 MPa, ~15 °C; 1 min	Approximately 10% inactivation after HPP. Further 30% inactivation after 58 days of refrigerated storage.	[96]
Cloudy apple juice (pH 3.5) from cv. Amasaya)	PPO 250–450 MPa, 25–50 °C, 0–60 min	~50% activity increase at 450 MPa, 25 °C, 15 min; 90% inactivation at 450 MPa, 50 °C, 60 min	[97]
Strawberry puree (cv. Aroma)	PPO, POD, 100–690 MPa, 24–90 °C, 5–15 min	~16% PPO activity increase at 690 MPa, 24 °C, 23% inactivation at 690 MPa and 90 °C POD inactivation at all conditions, almost complete inactivation after 5 min at 90 °C regardless of the pressure, 72% inactivation at 690 MPa, 24 °C, 15 min; slight antagonistic effect at P < 400 MPa	[98]
Strawberry puree	PPO, POD, 300 and 500 MPa, 0 and 50 °C, 1, 5, 15 min	72% and 50% inactivation of PPO and POD respectively at 500 Mpa, 50 °C for 15 min	[99]
Avocado paste	PPO 600 Mpa, 3 min	49.3% inactivation of PPO	[100]
Banana puree	PPO 517 and 689 MPa, 21 °C, 10 min	Increased activity after 10 min at 517 Mpa, 21% inactivation at 689 Mpa	[101]
Cantaloupe juice	POD, PPO 500 MPa, 20 min	22% and 91% inactivation of POD and PPO respectively	[102]
Carrot juice	POD, PPO 450 MPa and 600 MPa, 22 °C, 5 min	20% and 42% inactivation of POD and PPO, respectively, at 450 MPa; 30% and 31% inactivation of POD and PPO, respectively, at 600 MPa	[103]
White grape must	PPO 400–800 MPa, 25 °C, kinetic	Threshold of inactivation 600 MPa, 14% inactivation after 15 min at 800 MPa and 25 °C	[104]
Mango puree (pH 4.5) Puree with 200 ppm L-cysteine Puree with 500 ppm L-ascorbic acid	PPO 379–586 MPa, 25 °C, 0.03–20 min,	25% inactivation at all conditions after 20 min 95% inactivation at all conditions At P ≥ 448 MPa and time ≥ 10 min: 92% inactivation	[105]
Mango nectar (prepared from steam blanched mango slices with added sugar, citric acid (pH 3.95) and sodium erythorbate	PPO, POD, 600 MPa, 1 min	Complete inactivation of PPO and POD perhaps due to synergy with the steam blanching step	[106]
Plum puree	PPO 300, 600, 900 MPa Initial temperatures 60, 70, 80 °C, 1 min	~50% inactivation at 900 MPa and 50 °C and ~40% inactivation at 600 MPa regardless of the temperature	[107]
Plum puree (cv. ‘Sonogold’) Untreated Blanched	PPO 400 and 600 MPa, initial temperature 10 °C, 7 min	40% and 33% in the untreated samples and 35% and 15% increase in PPO activity in the blanched samples after treatment at 400 MPa and 600 MPa respectively	[108]
Açaí fruit	PPO, POD 400, 500, 600 MPa 25 °C and 65 °C, 5 and 15 min	PPO: 127% activity at 600 MPa for 5 min at 25 °C; 50% residual activity at 600 MPa for 5 min at 65 °C. POD: 100% activity at 600 MPa for 5 min at 25 °C; 117% activity at 600 MPa for 5 min at 65 °C.	[109]
Coconut water	200, 400 and 600 MPa, 40–90 °C, 1–30 min	33% and 22% inactivation of PPO and POD, respectively, at 600 MPa and 40 °C for 5 min; 80% and 85% inactivation of PPO and POD, respectively, at 600 MPa and 80 °C for 5 min	[110]

**Table 5 foods-11-00223-t005:** Summary of relevant studies on the effect of HPP on flavor modifying enzymes.

Product	Enzyme and Conditions	Results	Reference
Tomato dices	LOX 400, 600, 800 MPa, 25 and 45 °C, 1 and 5 min, tomato dices had pH in the range 4.36–4.42	Complete inactivation of LOX after 5 min treatment at 600; MPa or 1 min treatment at 800 MPa	[50]
Carrot juice	LOX 100–600 MPa, 25 °C, 10 min 300–500 MPa, 50–70 °C, 10 min Carrot juice was adjusted with Citric acid to pH 4.0.	83% inactivation of LOX at 300 MPa and ~65% inactivation at higher pressures. Complete inactivation of LOX at 500 MPa and 60 to 70 °C	[63]
Tomato pieces	LOX 0.1–500 MPa, −26 to 20 °C, 13 min As enzyme source the tomato blend with pH adjusted to 3.0 was used.	No inactivation of LOX at 20 °C and pressure less than 500 MPa, ~10% inactivation at 500 MPa and 20 °C, complete inactivation at −20 to −10 °C at 400 to 500 MPa, limited effect at 26 °C and pressures up to 500 MPa.	[94]
Avocado paste	LOX, 600 MPa, 3 min The pH was changed during storage for 58 days from 6.6 to 5.3.	44.9% inactivation of LOX	[100]
Cantaloupe juice	LOX, 500 MPa, 20 min, juice had pH in range 5.6–5.8.	95% inactivation of LOX	[102]
Tomato juice (4 °Brix)	LOX 100–650 MPa, 20 °C, 12 min enzyme activity was measured at pH 6.5.	Activity increase up to 400 MPa, complete inactivation at 550 MPa, 20 °C, 12 min	[77]
Green peas juice Whole green peas	LOX 0.1–625 MPa, -15–70 °C, kinetic; LOX activity was predicted as a pH function in the range 5.8–8.0	Antagonistic effects at pressures ≤650 MPa and temperature between -10 and 10 °C and pressures ≤200 MPa and temperature ≥60 °C, synergistic effects at other conditions; ~10% and 33% inactivation in juice at 500 MPa, 20 °C, 3 min and 500 MPa, 60 °C, 3 min compared to ~32% and 37% inactivation under the same condition in whole green peas (estimated based on the kinetic data)	[79]
Whole green beans Green beans juice Green beans juice	LOX 0.1–650 MPa, −10–70 °C, kinetic LOX activity was predicted at phosphate buffer (10 mM; pH 6). 500 MPa, 20 °C, 10 min LOX activity was determined at of air-saturated phosphate buffer solution (0.01 M; pH 6)	Antagonistic effects at pressures higher than 400 MPa and temperatures between −10 °C and 10 °C; lower stability in whole green beans 50% inactivation at 500 MPa and 20 °C for 10 min	[80,111]

### 4.3. Concluding Remarks

From the foregoing discussions, it is clear that high pressure processing specially around ambient temperature has limited impact on most quality degrading plant enzymes under commercially feasible processing condition (500 to 600 MPa, 3 to 5 min). This is more so for PPO, POD and PME whereas LOX and PG are relatively more pressure labile. Moreover, a significant increase in the activity of enzymes such as PPO, POD and PME occurs after high pressure processing of tissue systems (fruit and vegetable slices, purees, fruits) mainly due to the release of membrane bound enzymes as well as changes protein structure and conformation. Enzymes are often more pressure resistant than vegetative microorganisms and the residual activity of quality degrading enzymes may limit the shelf-life of high pressure processed products. This can be mitigated to a certain extent through modification of the physicochemical properties of the food matrix (lower pH, enzyme inhibiters and oxygen scavengers), oxygen exclusion and refrigerated storage. High pressure processing at elevated temperature improves the level of inactivation in many cases, although that may come at a cost in terms of quality retention. It has to be noted that the retention and activation of enzymes such as PME can be beneficial in applications such as improving the rheological and textural properties of some fruit and vegetable products as well as for creating novel structures.

## 5. Current Status of Industrial HPP Equipment for Food Processing

### 5.1. Short Historical Overview of Industrial High Pressure Processing Equipment

The first commercial high pressure processed product ever launched was Meidi Ya’s (Japan) “High Pressure’s Fruit Jelly” back in 1990 using a 30 L vessel, semi-industrial HPP unit manufactured in Japan by Kobelco (vessel and HPP machine) and Mitsubishi Heavy Industries (high pressure pumps). The next evolution in high pressure systems came in 1997 when Avomex—Fresherized Foods (USA) commissioned its first 35 L vessel system from the manufacturer Avure (formerly Flow). A third contender willing to take on this increasingly promising technology market was Alstom (France) who installed its first working industrial system in 1998 at the meat company Espuña (Spain). In this case a unit with a large 300 L vessel. Contemporarily, Avure was also manufacturing and installing their first 215 L vessels. The race towards larger, more productive HPP equipment had begun (Figure 1). Ting [112] defined fundamentals that any high pressure treatment machinery for food should fulfil.

Hyperbaric (Burgos, Spain) started its activity a bit later, in 1999, and installed their first 300 L vessel machines in 2002 at the meat industry Campofrío (Spain). In the past 20 years the competition has mainly taken place between Avure and Hyperbaric, who combined, have about 90% of all the industrial high pressure machines in production across the world. The lasting 10% market share belongs to few other equipment manufacturing companies in Asia or Europe. As of December 2019, the largest high pressure machine models have been installed. For HPP machine vendors building large volume machines (with long vessel and large diameter) require more know-how in designing vessel and frame (or yoke) to resist to pressure along the time, as a production machine should run reliably more than 10 years and 200,000 cycles at 6000 bar. As an example, if the vessel internal diameter is 40 cm, the high pressure plug area is 1256 cm^2^. As 6000 bar is 6000 kg/cm^2^, or 6 tons/cm^2^, force to support by each vessel plug is over 7500 tons. On one side, force on plugs (to be withstand by yoke) increases exponentially with vessel diameter. With a large vessel diameter like 40 cm, the considerable force generates quickly metal parts fatigue if design is not optimum. On the other side, long machines, accommodating high vessel volume, require excellent design to avoid yoke deformation and misalignment between vessel, plugs and yoke, which could cause a dramatic failure under pressure. This is why the size and diameter of HPP models have increasing slowly during the past, with the increasing knowledge of the manufacturers. An informative overview of principles and industrial applications was provided by Nguyen and Balasubramaniam [113] and recently by Barbosa-Cánovas et al. [114]. Any application in industry for specific food product must precede research about the pressure influence on microbial contamination. Black et al. [115] provided a good overview on this topic.

### 5.2. Evolution of the Adoption of the Technology

It has been indicated that the very first commercial system started operating in Japan back in 1990. A slow, very progressive adoption took place by pioneering food and beverage processors mainly in North America and Europe during the 1990s and early 2000s. As reflected in Figure 5, it took almost 20 years for the whole industry (all suppliers combined) to reach the milestone of HPP Machine number 100 (in 2008), whereas it took only 5 years more to reach Machine number 200 (2013) and it took less than 4 years (2017) to reach 400 HPP machines in production worldwide. At the end of 2019, the number of HPP units reached is about 520. A useful overview of the technology evolution was presented by Elamin et al. [116].

In recent years, more than 50 new units are being installed each year globally and the food industry is investing preferentially in large units. The main barriers for implementation of the technique were related to the high capital investment initially required, plus the relatively low throughput and volumes of the machines, and no less important the very maintenance-intensive environment that working cycles at 6000 bars created, which winged to meaningful aspects such as downtimes and spare parts costs. It is interesting how companies can utilize existing digitalization for promotion of high pressure technology (see [117]).

### 5.3. A Few Basic Concepts on HPP Systems

At the end of 2020, there were more than 590 industrial high pressure processing units running in the world, about twice than 5 years ago. About 20 of those industrial size systems (from 35 L up to 135 L vessel volume) are used for research and development purposes in industrial or academic laboratories and pilot plants. Thus, 570 units are dedicated to commercial food products’ high pressure processing.

The global major suppliers of HPP machines are Hiperbaric S.A. (https://www.hiperbaric.com, accessed on 28 August 2021, Burgos, Spain), which has some 280 industrial HPP units in production to date; JBT (https://www.avure-hpp-foods.com, accessed on 28 August 2021, Middletown, OH, USA), who it is estimated to have some 180 units currently in industrial production; Uhde High Pressure Technologies GmbH (https://www.uhde-hpt.com, accessed on 28 August 2021, Hagen, Germany), who have sold about 10 HPP systems. There are about 10 different Asiatic (Chinese, Korean and Japanese) brands or companies selling HPP machines in Asia where they have only collectively installed around 50 units to date. BaoTou Kefa High Pressure Technologies (https://www.btkf.com, accessed on 28 August 2021, Baotou, China) is the main Asiatic provider of HPP units having sold and installed about half of those 50 units.

As shown in Figure 2, North America has historically been, and continues to be, the main adopter and biggest driver for industrial HPP techniques for food and beverages (around 235 systems installed in total, with about 155 units in USA, 50 in Mexico and 30 in Canada), followed by the European Union with some 130 systems installed (the European leading country being Spain with 30 units); and Asia where almost 100 units are in production, mainly in China, South Korea, Japan and Thailand (totaling 70 HPP units, about half of them in China); Oceania (about 25 systems are in production in Australia and New Zealand) and South America (around 25 systems). In Africa, the only country running industrial HPP units is South Africa with eight units. All reflected figures are by the end of 2019. The largest and most productive systems have been launched in 2013 by Hiperbaric and Avure, with high pressures vessels of 525 L capacity, able to perform 10 cycles/h a 600 MPa with a holding time of 3 min. This allows throughputs over 3000 kg or liters of packaged food or beverages per hour (with a 525 L vessel filling efficiency of 60% corresponding to 315 kg or L of product processed per cycle). In the last 10 years, evolution of materials and designs has rendered possible that the HPP machines of today, with a similar level of investment, are up to 300% more productive than previous generations, and exponentially more reliable and efficient. This has had a very positive impact in the resulting costs per kg or liter of product being processed, and in the evolution of the adoption of this non-thermal technique. For example, prior to 2002, food was processed in HPP vessels vertically, where the product was loaded and unloaded inefficiently by the top of the vessel. In 2002, the first Hiperbaric horizontal HPP machine was installed in Spain in which food was introduced into the HPP vessel by one side and unloaded by the other side of the machine. This allowed a much better traceability, resulting in cost savings and efficiencies.

This part of the paper reviews the current status of the industrial offer of high pressure processing machines, the history perspective of its evolution, and the forecasts of growth and upcoming implementation, together with figures and current techno-economical estimates around HPP in the global food sector.

We will showcase and explain the different capacities and features of industrial high pressure equipment. A first common specification of these systems is that, up to the end of 2019, that they were all batch or in-pack HPP machines (processing pre-packed products), and there was no industrial semi-continuous or continuous HPP units processing liquids in-bulk (before packaging) in commercial production around the world given that the few semi-continuous HPP systems (by Avure formerly Flow) experimented for juice production in the late 1990 and were all stopped and dismantled in the early 2000′s. This has just changed, as in 2019 Hiperbaric installed for commercial juice production in a co-packer juice company (Atelier Hermes Boissons) in France its new technology, known as Hiperbaric Bulk, for processing liquids in-bulk, a unique model(to be expanded on later).

All the existing HPP systems (in-pack and in-bulk) work is based in what is commonly referred to as cycles (pressure cycles). For the industry, one cycle is the complete batch being high pressure processed, from the moment the target product is loaded into the high pressure machine, to the moment it is offloaded after being subject to certain pressure conditions during the designated amount of time. A “6000 bar/600 MPa/87,000 psi—3 min cycle” is typical nowadays in the food industry for many applications (e.g., guacamole, juices, cooked meats or ready to eat meals).

In a classical in-pack HPP unit, a cycle means loading all those packaged products into baskets or canisters pushed into the HPP chamber (also called high pressure vessel); filling that vessel with water, so all the products are submerged in that fluid; pumping more water into that chamber volume, hence building the isostatic pressure, in this case up to 6000 bar; holding that pressure up for a time from few seconds up to 10 min (depending product and shelf-life or pathogen inactivation to achieve); depressurizing, by evacuating from the vessel that extra water that had been pumped in; opening the chamber; pushing out of the chamber the baskets and offloading the processed product from the baskets (Figure 3). Twenty five years ago this complete cycle of 3 min at 6000 bar would have taken nearly 20 min to be completed. It took less than 10 min to be completed a decade ago; as of 2019, a typical, competitive HPP system would perform this job in 6 min.

Hiperbaric has developed a new “HPP Bulk” technology, which allows processing of beverages in-bulk before bottling. It works with a large a reusable and recyclable plastic flexible pouch or bladder inside the vessel and a system for loading and unloading liquids to process through a cleanable and sterilizable high pressure plug (Figure 4). After pressurized in the large pouch, the liquid is unloaded into an ESL (ultraclean extended shelf life) tank and goes directly into an ESL filling line for bottling in any kind of packaging material (cans, carton-brick, glass bottles, etc.).

The main advantages of this technology are cost efficiency (reduction of total cost of ownership), energy savings and high productivity as the filling vessel efficiency is >90%, fully automated installation, and it allows a variety of packaging options, regardless of the material, design or size. Moreover, as the basic process (high hydrostatic pressure) in-bulk is the same than the in-pack used for more than 30 years, so it is not considered as a new process by food safety regulators. The main limitation of this new technology is that an in-bulk machine can process only liquids (almost 500 L per cycle), but no solid or thick/viscous products. Furthermore, the investment cost for the whole processing line is higher because ESL or aseptic tanks, lines, valves and packaging machines are required for clean packaging after HPP.

Other key factors for in-pack and in-bulk units are the speed of pressure build-up, which depend on installed power and will be intrinsically related to the performance of the so called high pressure intensifiers. These are the pressure multiplier units that are used to pump water into the vessel as to create the target isostatic pressure. Some of its basic features will be explained as these are correlated to productivity and associated costs.

Considering that discussions about pressure systems usually involve their volumes and productivity, the main features of these machines that will be discussed hereafter are the vessel volume (the volume of the processing chamber/cylinder), the number of cycles performed per hour for a “typical cycle” which is in general 3 min holding time at 6000 bar and the vessel filling efficiency. This last factor, only relevant for in-pack machines depends on the product (mainly shape of its packaging and head space volume) but also can vary slightly with the diameter of the vessel.

### 5.4. Food Segments Adopting Industrial High Pressure Processing

Coincidentally or not, the three pioneering users of high pressure processing are also the sectors that own a biggest share of the *HPP pie*: fruit-based products, avocado products and meat products. Results of pressure treatment of meat were presented e.g., by Hugas et al. [118]. Staff of the Meat and Poultry journal presented interesting application of high pressure treatment on packed products to avoid contamination by pathogenic microorganisms such as *Listeria*, *E. coli*.

Fruit-based products are a vast category and certainly that pioneering niche of fruit jellies (Japan, 1990s) did not become main stream. It is instead the juice segment, the one that has more widely embraced the HPP solution across time, and particularly in the 2010–2019 period where more than 30% of the HPP systems sold (over 150 units) where installed in juice manufacturing companies Moreover, juices are the largest volume products to process for many HPP tolling companies (offering pay-as-you-go high pressure processing services), which are owing a total of about 100 HPP units globally (see part 2 of this paper for more information on this topic). Some case studies presenting selected food examples are featured by Tonello [119]. There is a niche in the juice business that has created a lot of traction: the cold pressed juice category, using claims such as “Never Heated”, “Unpasteurized”, “Cold Pressurized” has established itself in all premium juice markets worldwide, and is a great adopter of high pressure processing for shelf life and safety. The interest in this new juice space, cold pressed and without thermal processing steps, is reflected by the fact that large corporations have invested or purchased brands of HPP users. This has been the case of the 2012 Starbucks operation for Evolution Fresh (Rancho Cucamonga CA, USA) or the 2015 operation of Coca-Cola investing heavily into Suja Juice (Oceanside, CA, USA).

Processed avocado pulp and guacamole has maintained, across the relatively brief history of industrial high pressure processing, its status as an ongoing, mainstream adopter of this solution. Many leading suppliers of avocado pulp paste, guacamole and other derivatives implemented HPP following the success story that Avomex—Fresherized Foods (acquired by Hormel Foods Group in 2011) began in 1995 in USA and Mexico. The adoption was steady and as of December 2019, more than 75 HPP machines operate in avocado processors in 12 countries (namely, the major avocado fruit producing countries, including Mexico and USA but also Peru, Chile, Brazil, Costa Rica, Guatemala, Dominican Republic, South Africa, Spain, Australia, and New Zealand). More than 40 avocado/guacamole companies own HPP machines, including Calavo, Verfruco or Simplot in USA and Mexico and Frutas Montosa or Avomix in Spain.

The processed meat industry, as the main HPP users, has gradually included the high pressure processing step as means to better controlling shelf life and safety of many products. This sector comes third after juices with a total vessel number installed of around 90 units, and the major user of 300 L plus (300, 350, 420 and 525 L) HPP vessels with HPP tollers. Users include big names such as Hormel, Jennie-O, Cargill, Shuanghui, Campofrio, West Liberty Foods, Perdue Farms, Columbus, Cooper Farms, Applegate Organic Farms, Maple Leaf, Sofina, Butterball, Brasil Foods, Espuña and Noel.

Seafood processors were the fourth largest group of pioneers in using high pressure processing. This sector generally uses much lower levels of pressure, in the 3000 bar range, than the ones used in all the segments related above which generally perform cycles of over 4000 bar and most commonly, of 6000 bar. In this case, the target is to assist to shellfish shucking, and to facilitate the extraction of meat from large crustaceans. The biggest single segment using HPP in seafood is the lobster processors in the East Coast of North America New Bedford, followed by the Louisiana (New Bedford, MA, USA) oyster processing industry. Companies equipped with large HPP machines or several units are Motivatit Seafoods, Clearwater, Prestige Oysters, Westmorlands Fisheries, Riverside Lobsters, Seafarers, Greenhead Lobster and Cinq Degrés Ouest. Together, there are about 40 high pressure systems working for the seafood sector worldwide.

Other food categories where HPP adoption has been growing in the last years are dips (especially hummus), baby foods, pet food (raw meat products) and ready-meals. Sorenson and Henchion [120] studied consumers’ cognitive structures related to ready meals treated by high pressure.

### 5.5. Evolution of HPP Equipment: Volume, Speed, Productivity and Reduction of Cost

The progress for machinery suppliers has always been trying to build bigger machines, faster machines, with lower downtime (due to maintenance stops) and smaller total cost of ownership (TCO).

Commencing with size (equipment volumes): total vessel volume installed in the world is approximately 140,000 L (Figure 5), which means we are approaching an average machine volume of 240 L (considering a total number of units in production close to 600), versus previous figures of around 200 L ten years ago or 150 L fifteen years ago, and it is increasing as HPP product number and volume is constantly increasing in the market.

Following with an overview analysis of the equipment speed, meaning the ability to complete 6000 bar pressure cycles, if we take as example a 300 L high pressure machine of the 2005 class, and considering performing a full batch at the above mentioned 600 MPa with a holding time of 2 min, plus including all the machine movements (loading and unloading of carriers, low pressure filling, depressurization etc.) would require 11 min in total. The same 300 L system but of a 2010 install, would require 6.7 min for the same job (a 39% improvement). A 300 L system of 2015 required roughly 6.1 min for completing such cycle and in 2019 it only requires 5.5 min (another 10% improvement if compared with 4 years ago, and total a 50% reduction of cycle times within 15 years).

This translates easily to improved productivities in a meaningful manner. With basically the same level of capital investment in the equipment, that 300 L machine of 2005 would be producing 750–900 kg or liters of packaged food per hour. The throughput figure grows to 1200–1300 kg or liters per hour as of 2010. It means that the capacity corresponds to 1500–1800 kg or liters per hour in a current version of the same machine. This is a 100% increase over the last 15 years.

Lastly, machine reliability unfolds into avoiding downtimes, higher unit availability and hence into productivity. Although maintaining HPP systems is still demanding, current uptimes are consistent and most food industries utilizing the technique run their systems with two or three shifts per day, averaging in general between 20,000 and 40,000+ pressure cycles per year.

When we mix the above data and translate them into costs per kg or per liter, we can see clearly how the overall technological advances have meant that HPP technology is now much more feasible to implement, progressively cheaper to run, and always more justifiable in terms of TCO, financial margins and return on investment. Market forecast valid for years 2015 to 2025 for high pressure treated food products is available in the book by Peerun et al. [121]. Export in the food process technology sector was predicted by Purroy [122].

### 5.6. Installing an Industrial High Pressure System

Entry-level systems for niche productions (equipped with a vessel about 50 L) are relatively compact equipment that fit into 20 feet containers and can be transported, downloaded and commissioned quickly and easily. A Hiperbaric 55 L system which would typically produces around 200 to 350 kg or liters of packaged product per hour is roughly 8 m long × 2.3 m wide × 2.2 m tall and weighs some 20 Tons. It would usually be positioned, assembled, commissioned and started-up, including training, in less 7 working days of work carried out by one field engineer.

Larger systems such as 300–525 L vessel systems require more meaningful structures and are transported generally in a wooden crate carrying the main body of the machine (varying between 30 and 60 Tons of weight) plus a number of 40 feet containers. Such large systems can be up to 19 m long × 4 m wide × 4 m tall and weigh up to 90 Tons in total. Typically, a team of 3 field engineers would require between 7 and 10 days to download, position, assemble, commission and start-up one of these machines.

Recently, Jung and Tonello-Samson [123] presented challenges and limitations of the high pressure technology in industrial conditions.

It is the key for a swift and smooth commissioning that all utilities are in place by the time the new machine arrives to the food factory. These are, namely: energy supply; water (tap water within certain parameters of hardness, chlorides, total dissolved solids etc.); hydraulic oil (for the pumps and intensifiers); compressed air (necessary for the functioning of a number of valves and actuators); cooling (to maintain the temperatures of the process water and of the hydraulic oil within the target specs); and internet (for equipment monitoring, teleservice, upgrading, troubleshooting etc.).

In addition, the Hiperbaric Bulk units requires connections to cleaning-in-place systems, vapor and oxygen peroxide supplies for its cleaning and disinfection in place of tanks, valves and pipes in contact with high pressure processed beverage before its bottling or packaging under ultra-clean conditions.

### 5.7. Servicing and Maintaining the HPP Machine

Equipment working every cycle, every shift, every day at pressures of 6000 bar is, unavoidably so far, relatively intensive in preventative and corrective maintenance. As highlighted in previous paragraphs, this was possibly one of the reasons of the initial slow adoption of high pressure processing. In earlier generation of HPP machines down times were significant and maintenance costs in parts and labor were one of the main factors of concern for food processors resorting to this technology.

Due to a positive, incremental evolution of materials and designs, the HPP machines of today can be considered as “clockwork” if and when the preventative maintenance plans provided by the suppliers are followed consistently and seriously.

Fortunately for the industry, and this has obviously been reflected in the evolution of adoption of the technique, the lifespan of all the wear parts and spare parts subject to high pressure and subject to fatigue, has increased tremendously. Some examples follow (provided by the machine supplier Hiperbaric), which take into account the proper conditions of maintenance and water quality relevant in reliability:

Brass ring seals in the plugs that close the high pressure vessel were lasting in average around 5000 cycles fifteen years ago, whereas today their average lifespan reaches 20,000 cycles.

High pressure tubing would have lasted, in average, 7000 cycles of usage fifteen years ago, versus a life of over 50,000 cycles nowadays.

The wear parts in the pressure relief systems would eventually last some 1000 cycles back in 2005, whereas in 2019 the average lifespan is approaching 3000 cycles.

The high pressure vessel, which is considered the most critical component because of its cost if a replacement is needed, has also seen its engineering and reliability greatly increased. In the beginning of the XXI century, vessels working at 6000 bar could last less than 50,000 pressure cycles. From 2005 onwards, the major suppliers start guaranteeing their vessels in 100,000 cycles. It is common today to receive a guaranteed vessel life of 200,000 cycles. However, with the park of machines around the world, with former generations of vessel materials and as a consequence, limited life, the current situation requires suppliers to hold spare vessels for quick replacement and minimization of down times when a vessel reaches its life limit.

Technological improvements as highlighted above are a reality but still, industrial high pressure machines require dedicated attention and servicing. Daily preventative maintenance operations, and periodical (weekly, or based on number of cycles) corrective maintenance operations to prevent leaks and down times, are compulsory. In general, these are all mechanical in operation and the machine suppliers train the local maintenance field engineers on these tasks, which include replacement of high pressure seals, poppets, check valves, relief valve components, high pressure tubing, plug rings and seals.

After amortization of the initial capital investment (from EUR 0.5 to EUR 3 million depending on vessel volume and number of intensifiers installed), maintenance costs usually come second in the TOC spreadsheets of the food and beverage users of HPP. However, reliability, availability and maintainability of the machines have increased substantially over the years, and this is one of the clear roots to its increased adoption worldwide.

### 5.8. Concluding Remarks

There were almost 600 HPP industrial units in the world at the end of 2020, and forecasts indicate that this technology will continue to experience significant growth along these coming years because of increasing consumer demands for safe and fresher minimally processed food. The main sectors implementing HPP solutions are fruit-based products, avocado and dip products, meat products and seafood. In the last three years, the main drivers for HPP have been the juice segment (cold pressed juices) and the copacking sector (HPP tolling or contract processing). North America (USA, Mexico and Canada) has always been, and currently is, the main adopter and biggest driver for industrial HPP technique for food and beverages, followed by the EU, China and Korea, Oceania and South America.

Two suppliers Hiperbaric and Avure compete globally in this technology segment on a daily basis, and other prospect competitors, mainly from China, have been appearing in recent times. The largest and most productive in-pack systems have started to be installed in 2015 by Hiperbaric and Avure, with high pressure vessels of 525 L capacity, able to obtain throughputs of 2500+ kg or liters of packaged food or beverages per hour (considering 3 min holding time at 6000 bar and a 50% vessel filling efficiency). The in-pack HPP machines of today, with a similar level of investment than over 15 years ago, are up to twice more productive than previous generations, and exponentially more reliable and efficient. High pressure machines are still relatively maintenance-intensive, but their reliability and uptimes have increased dramatically over the last 15 years. This has contributed to the growth of the technology.

The very recent HPP equipment improvement is the unique Hiperbaric Bulk technology launched in 2019 for beverage processing. This fully automatic machine is a breakthrough innovation (patents pending) that roughly doubles vessel filling efficiency comparing to the same vessel volume in-pack machine, for an investment cost only 20% superior. It permits significant productivity increase (reaching up to 5000 L/vessel of 525 L) and a reduction by almost 50% of cost/L. Moreover, it enables the use of non-plastic packaging alternatives (cans, carton-bricks, glass, etc.). This will have a positive impact in the resulting costs per liter of liquids being processed, which is critical for HPP products to becoming mainstream and in the evolution of this non-thermal technique.

## 6. Conclusions

High pressure technology can be presented as an efficient industrial-scale preservation process that can retain many sensory and nutritional attributes of foods. We presented categories of commercially accessible fruit and vegetable products and a detailed methodology of HPP with the aim to inactivate food microorganisms. This review also discussed challenges with endogenous enzymes present in fruit and vegetable raw materials and their inactivation by HPP. Future research on HPP of foods will need to further the scientific evidence on the nutritional benefits of the technology to the consumer. Another limitation of HPP is its relative capital and energy intensive nature. Thus, adaptation of the latest material science and design as well as innovations to improve utilization and energy efficiency of the process can ensure wider adaptation and future sustainability of HPP.

## Figures and Tables

**Figure 1 foods-11-00223-f001:**
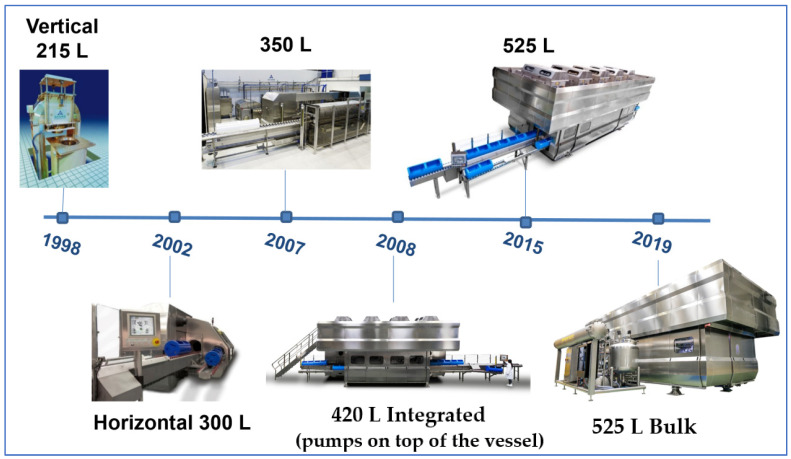
Evolution of HPP equipment size and volume over the last 23 years.

**Figure 2 foods-11-00223-f002:**
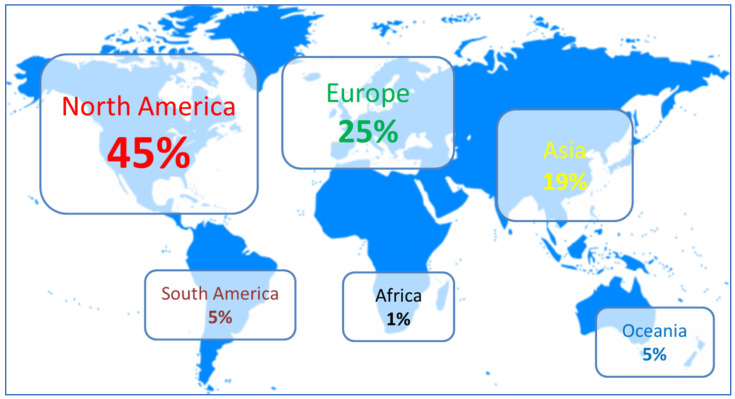
Industrial implementation of HPP technology by geographical region, as of 2019 (almost 600 industrial HPP machines run in production globally).

**Figure 3 foods-11-00223-f003:**
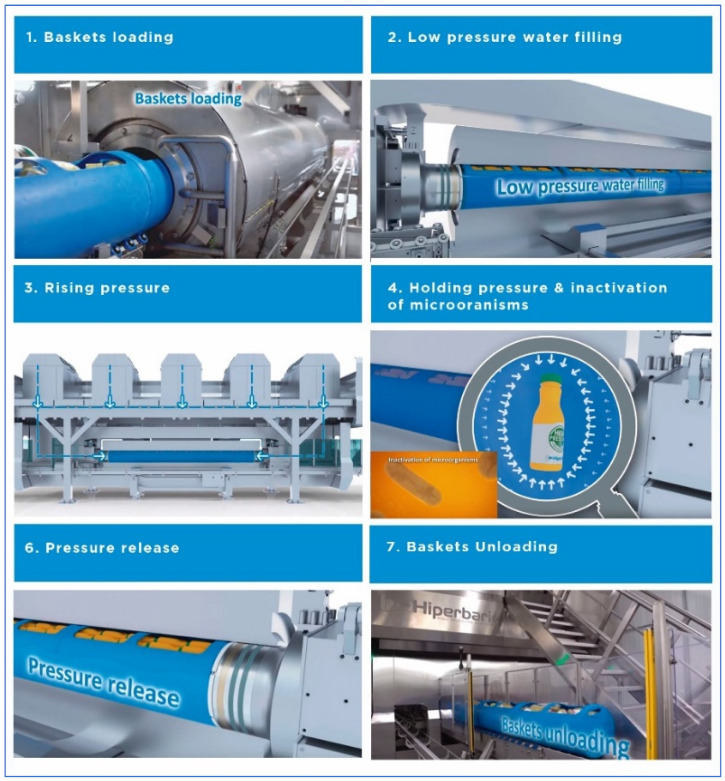
HPP in-pack processing sequence.

**Figure 4 foods-11-00223-f004:**
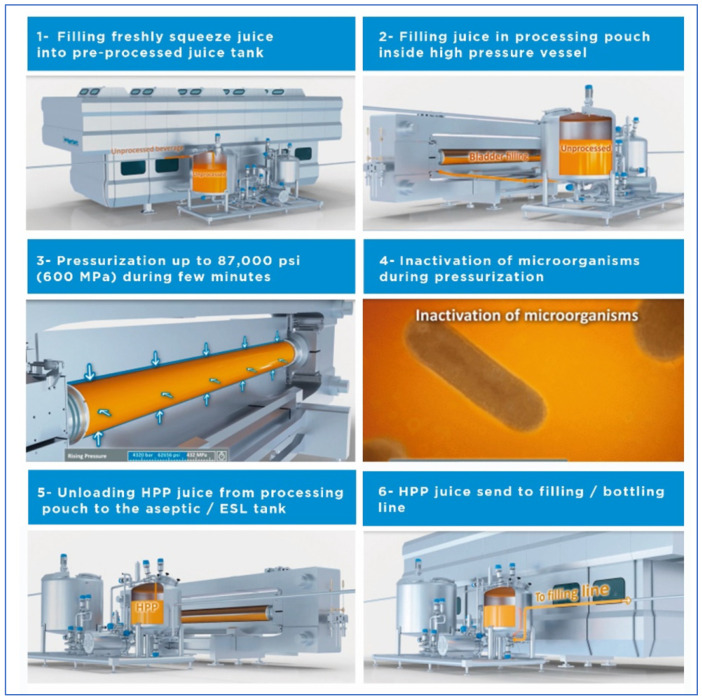
HPP in-bulk processing sequence.

**Figure 5 foods-11-00223-f005:**
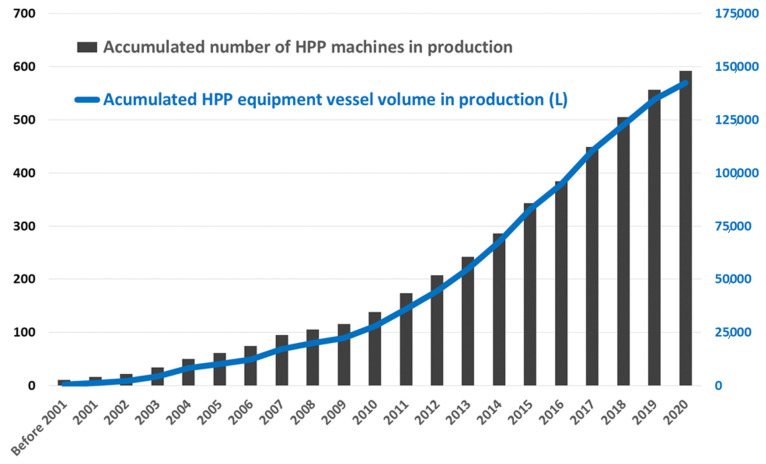
Evolution of number of industrial high pressure processing machines in production across time, accumulated and total volume of vessel (Hiperbaric and all others HPP machinery vendors).

**Table 1 foods-11-00223-t001:** Models used to describe different microbes inactivation in fruit products after HPP and HPTP.

	Fruit Products	Soluble Solids (°Brix)	Model	Pressure (MPa)	Temperature (°C)	Model Parameters *	Reference
**Spores of bacteria**							
*Alicyclobacillus acidoterrestris* NZRM 4447 (ATCC 49025)	Apple juice Lime juice conc. Blackcurrant juice concentrate	10.6 20.2 30.3	First order	600	45	*D*-value: 8.6 min 19.9 min 46.1 min	Uchida and Silva [12]
*Bacillus coagulans185A*	Tomato juice	nr	Weibull	600	95	*b* = 1.93; *n* = 0.68	Daryaei and Balasubramaniam [13]
*Bacillus coagulans* ATCC 7050	Tomato pulp	4.0	Biphasic	600	60	*D*-value ^a^: (1) 1.6 min; (2) 6.2 min	Zimmermann et al. [14]
**Spores of molds**							
*Byssochlamys nivea* JCM 12806 (CBS 696.95)	Strawberry puree	8.1	Weibull	600	75	*b* = 0.29; *n* = 0.66	Evelyn and Silva [15]
*Neosartorya fischeri* JCM 1740 (ATCC 1020)	Apple juice	10.6	Weibull	600	75	*b* = 1.44; *n* = 0.35	Evelyn et al. [16]
*Eurotium repens* DSMZ 62631	Apple juice	12.4	Biphasic	500	45	*D*-value ^a^: (1) 2.0 min; (2) 9.0 min	Merkulow et al. [17]
**Spores of yeasts**							
*Saccharomyces cerevisiae*	Orange juice	11	First order	500	Room T	*D*-value: 0.067 min; *z_P_*-value: 123 MPa	Parish [18]
**Vegetative bacteria**							
*Leuconostoc mesenteroides* ATCC 8293	Orange juice Orange juice conc.	11.4 42	First order	350 400	Room T	*D*-value: 2.0 min; *z_P_*-value: 137 MPa *D*-value: 6.1 min; *z_P_*-value: 251 MPa	Basak et al. [19]
*Lactobacillus brevis*	Orange juice	11.6	First order	350	Room T	*D*-value: 0.67 min; *z_P_*-value: 105 MPa	Katsaros et al. [20]
**Vegetative yeasts**							
*Zygosaccharomyces**bailii* ATCC 2333	Mango juice	15	First order	350	Room T	*D*-value: 0.62 min; *z_P_*-value: 84 MPa	Hiremath and Ramaswamy [21]
*Saccharomyces cerevisiae*	Orange juice Orange juice conc.	11.4 42	First order	250 400	Room T	*D*-value: 5.4 min; *z_P_*-value: 135 MPa *D*-value: 23.5 min; *z_P_*-value: 287 MPa	Basak et al. [19]
ATCC 38618							

* *D*- and *z*-values are the first order kinetic parameters; *b* and *n* are the Weibull scale and shape factors (*log N*/*N*_0_ = −*bt^n^*), respectively; nr—not reported. ^a^ The biphasic model assumes two rates of inactivation corresponding to two *D*-values. The *D*-values were calculated from the inactivation rates published.

**Table 2 foods-11-00223-t002:** Modeling the microbial inactivation in vegetable products after HPP and high pressure thermal processing (HPTP) *.

Form	Vegetable Products	Model	Pressure (MPa)	Temperature (°C)	Kinetic Parameters	Reference
**Spores**						
*Bacillus licheniformis*	Carrot juice	First order	600	60	*D*-value: 0.70 min *z*_P_-value = 339 MPa; *z*_T_-value= 23.2 °C	Tola and Ramaswamy [25]
		Weibull	600	60	*b* = 0.13; *n* =0.70	
*Eurotium repens* DSMZ 62631	Broccoli juice	Biphasic	500	45	*D*-value: (1) 16.0 min; (2) no inactivation	Merkulow et al. [17]
*Penicillium expansum* DSMZ 1994 (CECT 2279)	Broccoli juice	Biphasic	350	40	*D*-value: (1) <1 min; (2) no inactivation	Merkulow et al. [17]
**Vegetative cells**						
*Escherichia coli* MG 1655 (ATCC 47076)	Carrot juice	First order	600	20	*D*-value: 2.5 min	Van Opstal et al. [26]

* *D*- and *z*-values are the first order kinetic parameters; *b* and *n* are the scale and shape factors of the Weibull model, respectively.

## Data Availability

All data files are available at paper authors.

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
