# Peer review of "High Pressure Processing Applications in Plant Foods"

_foods, 2022, doi:10.3390/foods11020223_

Round 1
Reviewer 1 Report
This review paper shows applications of high pressure processing on plant foods incuding inactivation of microbes and enzymes in fruits and vegetables in Chap 3 and 4. This review paper also shows HPP machine, history in Chapter 5. Chapter 4 was well written, while chapter 3 and 5 needs lots modifications. According to author contributions, authors wrote chapters individually, then reviewer think that the first author had not editted the texts all over. Some texts and contents were in duplicate within the single review paper. Additionally, some advertise-like texts were shown in Chapter 5. Delete such texts.
So, authors need to polish and shorten the manuscript.
Some contents were written several times, therefore authors totally modifiy the text after no duplicate. I recommend Major revision.
In chapter 2 and Tables A1-18(18 must be 16) in Lines 50-59, there were lots of HPP products.
Similar industrial products were shown in chapter 5.
The title of this review paper was focusing on Plant foods, but also seafood were written in chapter 5. Therefore, authors have to change the title of this review paper.
L50 Although authors wrote tables A1-18, there were only 16 tables in appendix B in L.880 p.32 to L.922 in P.36.
L50 "Tables A1 up to A18, see Appendix A," should be A16, see Appendix B
Or L.880, Appendix B should be changed to A.
In chapter 3 in Lines 66 to 208, formatting of manuscripts were not correct. For example,
author names were not essential in the text, it was because that reference numbers were
enought to know reference details. In fact, authors write the reference number only in Tables
3, 4 and 5 in chapter 4.
Combinations of pressure level, pressuring time and temperatures were essential for this
kind of review paper to compare the results.
L80: Delete , from [5,].
L140 Unit of temperature shoud be correctly modified.
In chapter 4, abbreviation HPTP should be used in L.284 similar to L.79.
L.329 PPO was abbreviation of polyphenol oxidase. Then peroxidase should be POD.
Authors wrote PPO for both polyphenol oxidase and peroxidase.
L.367 Order of reference numbers need modification. 53==>99
L.392 Delete and - from [reference nuber].
L.491 Delete , from [] and modify the order of references.
Tables 3 p14, Watermelon should be Watermelon juice from ref 82.
Tables 3 p15, Delete P= at ref.101.
Add tthe combinations of pressure level, pressuring time and temperatures in tables.
Tables 4 p17 Apricot nectars, 5-20 must be pressuring time, then add min after 5-20 from ref 91.
Tables 4 p20, similar to mentioned above, watermelon should be watermelon juice from ref 82.
In chapter 5, the explanation about Hiperbaric products were too much and too long.
Shorten the texts from L.549-576, and L.614-643.
5.2 short history and 5.3 in L.645-662 should be added to 5.1 L.470.
L.600 Mitsubishi should be Mitsubishi Heavy Industires.
L.893 Font size of caption in tableA6 needs modification.
References;
In journal names, "and " and "&" were used. Especially IFSET should be correctly modified.
L.1129 and L.1197 Italic character for scientific name.
L.1193 ref 130 would be the same as ref 60.
Reviewer hopes these suggestions above will help author's modification for publication.
Author Response
Replies are given in attached file.
Milan Houška

Reviewer 2 Report
In general, the whole article is very well organized and written. The graphics resolution must be improved. I also have recommended some minor changes to improve this paper:
General comments: this article reviews High Pressure applications in plant foods. The revision period should be mentioned in the abstract or in the introduction. An index should be inserted to give a first overview of the topics reviewed. I miss a part containing more aspects on pilot plant equipment, and the issues related to scale up.
In general, the whole article is very well organized and written. I only recommend some minor changes to improve it.
Abstract:
Lines
- Vitamins and minerals are nutrients and should be included inside this concept. What do
you mean with health effective components?
- Introduction:
Lines
27 Please, add an index before this section.
40 I also recommend to mention the period reviewed (year range).
- Food products of fruit and vegetable origin
Lines
50 It seems that the factories manufacturing HPP treated products are only the ones with
installed Hiperbaric equipment. You can check and update other companies reported by
other authors as appearing, for example, in Daher et al., 2017 (doi: 10.3390/agriculture7090072)
- Effects of HPP and heat on key spoilage/pathogenic microorganisms in fruit and vegetable products
Lines
Mentioning legal criteria (USA/EU) related to microbial inactivation should improve the paper.
104 Before mentioning D, N, b and zP values, these parameters should be briefly described. Accordingly, the footnotes in Tables 1 and 2 would be better inserted in the text. In this way, inactivation parameters as b and D values would be briefly explained.
135 It seems that pH is < 4.6 instead pH > 4.6
145 A sentence reporting that HPP lows the pH should be mentioned.
- The effect of high pressure on endogenous fruit and vegetable enzymes
Lines
237-46 I wonder if all of these modifications happen in all enzymes, please refer to the enzyme or
types of enzymes reported by the cited authors.
Conclusions:
The authors should report something related to the future perspectives for this technology.
Tables and Figures:
Contents in Tables A1 to A8 should be ordered by one criterium, for example alphabetically.
Table 1: include the word Form in the first row, similarly as in Table 2.
I recommend adding the respective equations used in these 2 tables to obtain D and b values, by adding a new column in both tables 1 and 2.
Table 2 legend is in page 6 instead page 7.
Table 3: 3rd row and 2nd column, the sentence is not ended. 9th row and 2nd column, suppress the word enzyme. This Table should be ordered by a criteria as alphabetical order of pressurized products.
Tables A4, A5, A6 and A8 do not contain products images/pictures. Should be possible to add them?
In Table 4, add min after 5-20 in row 7, for apricot nectars. In addition, other 3 rows do not report processing time (see rows related to orange juice, Greek navel orange juice and Florida orange juice).
In Table 5, first row, substitute the character / by a comma. Some rows do not report the treatment time (see tomato dices in Table 5)
All the tables should present the same order and format for treatment conditions. Please check all of them.
In the Figures 2 and 3, the text needs a better resolution to facilitate the reading.
References:
Authors should follow author guides to write the references in the format required by Foods.
Some Authors surnames should be suppressed before the respective cites along the whole text.
Cited Journals should be abbreviated.
DOIs should be added to the cites.
In line 978, journal should be in italics.
In most of the references a comma should be inserted between surnames and initial names.
Missprints and format errors:
In line 135, I suppose that pH should be < 4.6 instead >4.6.
In line 141, suppress the comma before [22].
In lines 160, 643, et al. should be typed in italics.
In line 184, parenthesis in pressure should be typed in the same line as pressure.
In line 257, type this subsection in italics and without bold.
In line 351, type physicochemical instead physiochemical.
Reviewer 3 Report
It is difficult to see the novelty of the manuscript since the topic of HPP has been reviewed extensively in both manuscripts and book chapters. The manuscript lacks organization – at times the authors jumped over the place without following a logical order, making it difficult to read.
- Line 14: what the authors mean by pre-packed?
- Line 15: what the authors mean by homogenate products?
- Line 18: what is the difference between vitamins, nutrients, and health effective components? Please define what are those health effective components
- Line 42: I do not agree with the term gentle preservation. There is no such thing as gentle processing in terms of processing. I do believe it causes confusion due to its ambiguity.
- Line 94: was linear? Or it was thought to be linear?
- Line 101: is that maximum working temperature (room plus adiabatic heat)?
Author Response
Replies on your comments are given in the attached file.
